# Raman and Infrared Spectroscopy of Barium-Gallo Germanate Glasses Containing B_2_O_3_/TiO_2_

**DOI:** 10.3390/ma16041516

**Published:** 2023-02-11

**Authors:** Karolina Kowalska, Marta Kuwik, Joanna Pisarska, Maciej Sitarz, Wojciech A. Pisarski

**Affiliations:** 1Institute of Chemistry, University of Silesia, Szkolna 9 Street, 40-007 Katowice, Poland; 2Faculty of Materials Science and Ceramics, AGH University of Science and Technology, 30 Mickiewicza Av., 30-059 Kraków, Poland

**Keywords:** barium gallo-germanate glasses, boron trioxide, titanium dioxide, vibrational raman and infrared spectroscopies, structure–property relationship

## Abstract

Modified barium gallo-germanate glass hosts are still worthy of attention in studying structure–property relationships. In this work, two different series of glass systems based on (60-x)GeO_2_-xTiO_2_-30BaO-10Ga_2_O_3_ and (60-x)GeO_2_-xB_2_O_3_-30BaO-10Ga_2_O_3_ (x = 10, 30, 50 mol%) were synthesized, and their properties were studied using spectroscopic techniques. X-ray diffraction (XRD) patterns revealed that all fabricated glasses were fully amorphous material. The absorption edge shifted toward the longer wavelengths with a gradual substitution of GeO_2_. The spectroscopic assignments of titanium ions were performed with excitation and emission spectra compared to the additional sample containing an extremely low content of TiO_2_ (0.005 mol%). On the basis of Raman and FT-IR investigations, it was found that increasing the TiO_2_ content caused a destructive effect on the GeO_4_ and GeO_6_ structural units. The Raman spectra of a sample containing a predominantly TiO_2_ (50 mol%) proved that the band was located near 650 cm^−1^, which corresponded to the stretching vibration of Ti-O in TiO_6_ unit. The deconvoluted IR results showed that the germanate glass network consisted of the coexistence of two BO_3_ and BO_4_ structural groups. Based on the experimental investigations, we concluded that the developed materials are a promising candidate for use as novel glass host matrices for doping rare-earth and/or transition metal ions.

## 1. Introduction

Glass, according to the definition, is a supercooled liquid and exhibits an amorphous nature. It should be noted that the common specific feature of all oxide glasses is the lack of long-range ordering. In the last few years, much research has been focused on explaining the relationship between the content of network formers and network modifiers and the stretching or bending vibrations occurring in the glasses. Special attention has been paid to the attractive infrared range characterizing the local structure of glassy systems using vibration analysis such as Raman and FTIR spectroscopy. There are extremely useful experimental techniques in investigating the structural properties of inorganic glasses [1,2,3]. Although there are well-documented scientific reports, the interpretation of the vibrations of glass host structural units occurring in oxide glasses is still a subject of ongoing scientific debate. 

The above vibrational spectroscopies were successfully applied to examine the local structure of borate [4,5], phosphate [6,7], tellurite [8,9], silicate [10,11], and germanate [12,13,14,15,16] glass-host matrices. Studies in the last decade have shown that basic glass composition can be realized by adding several oxides and/or fluoride additives [17,18]. The main goal is to develop materials with excellent thermal, chemical, structural, and spectroscopic properties and various potential applications. Among those mentioned above, germanate glass is a promising starting material. First observed by Ivanov and Evstropiev [19], the coordination of Ge would continuously change from GeO_4_ to GeO_6_ with the addition of alkali oxides. Recently, the structural properties of germanate-based glasses have been examined in Na_2_O-GeO_2_-TeO_2_ [20], Li_2_O-GeO_2_-TeO_2_ [20], MnO-GeO_2_-PbO_2_ [21], TeO_2_-GeO_2_-PbO [22], Na_2_CO_3_-CaO-GeO_2_ [23], and Ga_2_O_3_-GeO_2_-BaO [24]. Among the reported germanate-based glasses, the barium gallo-germanate matrix has received the most attention due to its excellent properties, i.e., relatively large glass-forming region, high transmittance in a wide wavelength region, superior chemical durability, and thermal stability [25,26]. The structural properties of the BGG glass system modified by boron trioxide are less documented in the literature. Borate-based glass has excellent properties among the common glass-forming agents due to its high bond strength, coordination geometry, and maximum phonon energy (~1400 cm^−1^). The transformation of the coordination environment of boron, in terms of the equilibrium of the structural changes related to the conversion between BO_3_ and BO_4_, has been extensively studied [27]. Meanwhile, titanium oxide (TiO_2_) is one of the important constituents of luminescent glass, which acts as both network formers and network modifier [28]. Depending on the glass type, titanium can exhibit both trivalent and tetravalent valence states. The first attempts in this area were made by Cheng J. and Chen W. [29], who showed that stable titanate glasses could be obtained without adding glass network components (SiO_2_, B_2_O_3_). However, the formation of titanium ions in glasses has been reported, but most of the glass systems are partially crystallized. Up to now, fully amorphous barium gallo-germanate glasses containing higher TiO_2_ contents (10–50 mol%) were not obtained. It is well known that metal oxides such as B_2_O_3_ and TiO_2_ can be added to germanate-based glass to modify their optical and structural properties. The effect of the spectroscopic properties of properties for SrO-BaO-B_2_O_2_-SiO_2_ glass ceramics [30] with different TiO_2_/B_2_O_3_ ratios has been studied. Consequently, the fraction in the form of TiO_4_ and TiO_6_ increased when the fraction of BO_3_ decreased. For this reason, it is important to broaden our knowledge of the structural properties of the GeO_2_-BaO-Ga_2_O_3_ glass system modified by TiO_2_/B_2_O_3_.

In our previous work [31], Cr^3+^ ions were used in multicomponent germanate glasses as a spectroscopic probe. We observed that GeO_2_ and TiO_2_/B_2_O_3_ strongly influenced the profiles of luminescence bands and their relative intensity of trivalent chromium ions in the visible and infrared range. The calculated spectroscopic parameters from the Tanabe–Sugano diagram indicated that Cr^3+^ ions in the GeO_2_-BaO-Ga_2_O_3_ glassy phase occupied the intermediate (sample with B_2_O_3_) and strong (sample with TiO_2_) crystal field, respectively. It is assumed that adding glass formers such as TiO_2_ and B_2_O_3_ to the BGG network changes the local structure. We have extended this experimental approach to the analysis structural properties of undoped GeO_2_-TiO_2_-BaO-Ga_2_O_3_ and GeO-B_2_O_3_-BaO-Ga_2_O_3_ systems. Structural changes in the studied glasses were examined by XRD, FT-IR and Raman spectroscopy. Our study intends to present the role of titanium ions in germanate glass-host matrices based on excitation and emission spectra compared to the glass sample containing an extremely low content of TiO_2_. Another aspect worth exploring is a mathematical procedure of spectral decomposition as the possibility of determining the component bands resulting from matrix vibrations for the GeO_2_-B_2_O_3_-BaO-Ga_2_O_3_ glass system. 

## 2. Materials and Methods

In the presented procedure, multicomponent germanate-based glass systems modified by TiO_2_/B_2_O_3_ were prepared using the conventional melt quench technique. The high purity chemicals such as (>99.99%) germanium oxide (GeO_2_), boron trioxide (B_2_O_3_), titanium oxide (TiO_2_), barium oxide (BaO), and gallium trioxide (Ga_2_O_3_) (Sigma Aldrich Chemical Co., St. Louis, MO, USA) were used to synthesize the glass samples. The glass samples were entitled as GT1 (GeO_2_:TiO_2_ = 5:1), GT2 (GeO_2_:TiO_2_ = 1:1), GT3 (GeO_2_:TiO_2_ = 1:5) and GB1 (GeO_2_:B_2_O_3_ = 5:1), GB2 (GeO_2_:B_2_O_3_ = 1:1), and GB3 (GeO_2_:B_2_O_3_ = 1:5), where the first letters refer of the GeO_2_, TiO_2_, and B_2_O_3_ metal oxides. The chemical compositions of the GeO_2_/TiO_2_ and GeO_2_/B_2_O_3_ molar ratios and nomenclature are listed in Table 1 and Table 2. The glass components were taken in appropriate proportions and mixed using agate-made mortar and pestle in a glove box in an inert atmosphere to constitute a 5 g batch. The homogenized mixtures were melted in a crucible (Łukasiewicz Research Network, Institute of Ceramics and Building Materials, Cracow, Poland) at a temperature of 1250 °C for 60 min in an electric furnace (CZYLOK Company, Jastrzębie-Zdrój, Poland). The molten glass sample was taken out of the electric furnace, cast on a porcelain plate, and cooled down to room temperature. At the end of the procedure, the glass samples were polished (semiautomatic grinding and polishing LaboPol-5 Struers, Denmark) in the desired shape. Finally, we fabricated glass samples of dimensions 15 mm × 15 mm and thickness ±3 mm for the following optical and structural measurements. Figure 1 and Figure 2 show images of the polished samples GT2 and GB2.

An XRD spectrum for prepared samples was recorded with an X’Pert Pro diffractometer with CuKα radiation with a λ = 1.54056 Å wavelength supplied by PANalytical (Almelo, The Netherlands) in the range 20–70°. The diffraction patterns were measured in step-scan mode with a step size of 0.050 and a time per step of 10 s. The UV-VIS spectrophotometer (Varian Cary 5000, Agilent Technology, Santa Clara, CA, USA) was used to measure the optical absorption spectra. The excitation and luminescence spectra of the glasses in a range of 260–650 nm were registered using a laser system that consisted of a PTI Quanta-Master 40 UV/VIS Steady State Spectrofluorometer (Photon Technology International, Birmingham, NJ, USA) coupled with a tunable pulsed optical parametric oscillator (OPO) pumped by the third harmonic of an Nd:YAG laser (Opotek Opolette 355 LD, OPOTEK, Carlsbad, CA, USA). The laser system was coupled with a Xe lamp (75 W). The resolution for the excitation and luminescence spectra was ±0.25 nm.

In the next step, the structural investigations for the obtained materials were evaluated. The bonding vibrations were determined via a Fourier Transform Infrared (FTIR) measurement in the region 1600-400 cm^−1^ (Nicolet™ iS™ 50, Thermo Fisher Scientific, Waltham, MA, USA) with a diamond attenuated total reflectance (ATR) module. The complementary structural characterization of the obtained glass samples was verified using Raman spectroscopy (Thermo Scientific, Waltham, MA, USA). The appropriate laser source with an excitation wavelength of 780 nm was used to obtain the Raman spectra. The laser was directly focused on the sample with an Olympus long-working-distance microscope objective (50×). The Raman and IR spectra were normalized and deconvoluted using Origin Pro 9.1 software. All the measurements were performed at room temperature.

## 3. Results

The aim of this work is to evaluate the effect of a boron trioxide (B_2_O_3_) and titanium dioxide (TiO_2_) substitution on the properties of barium gallo-germanate (BGG) glasses. Firstly, to emphasize the potential of titanium-rich glasses and boron-rich BGG glasses, X-ray diffraction (XRD) was used to verify the local structure of the studied glass systems. Absorption spectroscopy is a very useful technique for characterizing the optical properties within the range of 200–800 nm of fabricated glasses. The assignment of the titanium ions’ emission bands that were derived from the spectroscopic results allowed us to confirm the Ti^3+^ ions in barium gallo-germanate glass systems. Secondly, a thorough analysis of the structural properties was performed by means of FT-IR and Raman spectroscopy. In those pioneering works, the properties of barium gallo-germanate systems were studied for possible applications in low-loss fiber optics and optical components [32,33,34]. The morphology of barium germanate glasses was reported by Shelby [35]. It should be noted that the barium gallo-germanate system shows a broad glass-forming region [36]. The properties of these systems can also be modified by adding or substituting other components. The effect of various substitutions in the barium gallo-germanate glasses was studied by Jewell et al. [37]. As a result, gadolinium is a typical modifier ion because of its large field strengths. Next, aluminum acts as an intermediate with AlO_4_^−^ substituting directly for Ga_2_O_4_^−^ units. In the presence of GeO_2_, gallium atoms tend to reinforce the glass network as observed in BaO-Ga_2_O_3_-GeO_2_ glass compositions, where the corners bind GaO_4_ and GeO_4_ tetrahedra. Consequently, depending on the content of various components, their role can change from network modifier to network former. To the best of our knowledge, these phenomena were not yet examined for B_2_O_3_, one of the typical covalent network formers that meet Zachariasen’s rules for glass formation, and TiO_2_, whose role can change from glass network modifier to glass network former depending on its amount in the glass.

### 3.1. Glass Characterization

In order to examine the amorphous or crystalline state of the fabricated glasses, a phase analysis was conducted with the use of X-ray diffraction (XRD). Figure 3a,b present representative X-ray diffraction patterns measured of the barium gallo-germanate glasses with the varying content of TiO_2_ and B_2_O_3_ (from 10 to 50 mol%). According to the literature data [38], the barium gallo-germanate glass system can exhibit a tendency toward surface crystallization. However, it is well known that bulk crystallization can be made possible in various glass systems by phase separation followed by nucleation and crystal growth. It needs to be stressed that TiO_2_ was an effective crystal nucleating agent in various glass systems. Systematic studies clearly indicate that the crystallization of TiO_2_ in glass systems depends on several factors, e.g., ionic size, ionic chance, and the ability to be conditional glass former ions [39,40]. The main drawbacks of TiO_2_ content are related to decreasing the viscosity of the glasses at a high temperature process and importantly promoting glass liquid–liquid phase separation, which provides more phase interfaces. Mingshen et al. [41] exhibited the role of TiO_2_ (5–10 wt%) in the process of the phase separation, nucleation, and crystallization of CaO-MgO-Al_2_O_3_-SiO_2_-Na_2_O system glasses. Importantly, they showed that the number of the main crystalline phase increased with temperature and time. Our results indicated that this phenomenon was not observed in the fabricated glass system using the traditional high-temperature melt-quenching method. For this reason, for the structural analysis, we started using X-ray diffraction measurements.

Figure 3 shows the XRD patterns of the GT and GB precursor glasses. The titanate–germanate samples revealed only a broad diffuse scattering at different angles instead of narrow lines typical for crystalline materials, confirming a long-range structural disorder characteristic of the amorphous glassy network. The same behavior was observed in the case of a glass sample where GeO_2_ was partially substituted by B_2_O_3_ from Figure 3b. The broad low intense peak at the angles 20–30° confirmed the amorphous nature of the GB1, GB2, and GB3 samples. Additionally, the hump’s maximum did not shift, corroborating the absence of the evolution to a lower degree of the order of the local structure of the studied glasses with wide GeO_2_:TiO_2_ and GeO_2_:B_2_O_3_ molar ratios. Moreover, the GT3 glass sample containing 50 mol% TiO_2_ showed a higher peak intensity than the glass sample with predominantly boron trioxide content GB3. This phenomenon met the requirement that the higher the atomic number of an element (Ti > B), the higher the X-ray diffraction intensity shown in the inset of Figure 3. 

Figure 4 illustrates the optical UV–visible absorption spectra for fabricated germanate glasses modified by titanium dioxide (Figure 4a) and boron trioxide (Figure 4b). It should be noted that the characteristic of both series of GeO_2_-TiO_2_-BaO-Ga_2_O_3_ and GeO_2_-B_2_O_3_-BaO-Ga_2_O_3_ glass matrices is the absorption edge which was shifting towards the longer wavelength with increasing TiO_2_ and B_2_O_3_ content. Following that, the UV cut-off wavelength, referred to as the intersection between the zero-base line and the extrapolation of the absorption edge, was estimated. The inset in Figure 4a,b presents the absorption edge for the GT1, GT2, GT3 and GB1, GB2, GB3 glasses. From absorption spectra, it is clear that with the increasing TiO_2_/B_2_O_3_ concentration, the absorption edges were shifted to longer wavelengths. 

According to the literature [42,43], titanium ions are also accepted to exist in glasses in various oxidation states. For this reason, we extended our experimental approach in optical research by designing and preparing a glass sample doped with an extremely low concentration of TiO_2_ (0.005 mol% TiO_2_). We registered the optical absorption spectrum for the spectroscopic characterization of titanium states in the glass sample (Figure 5a). Rao et al. [44] suggest that the intense and very broad absorption in the visible region was a result of superimposed bands from Ti^3+^ and Ti^4+^ ions. In the silica calcium aluminosilicate system, the band of about 310 nm corresponded to the Ti^4+^ ions [45]. However, this band was strongly masked by the absorption edge for titanium-doped germanate glass, which was significantly shifted toward longer wavelengths. Moreover, in the case of a fabricated sample doped with titanium ions, the inset of Figure 5a displays the high-resolution optical absorption spectrum in the 675–730 nm range. The low intense absorption band centered at 700 nm may be quite well interpreted and related to the Ti^3+^–Ti^4+^ ion pair interactions [45]. Based on it, it was stated that the presented experiment constitutes the next step toward measurements with different emission wavelengths

To analyze the emission spectra for the prepared glasses, it is necessary to know the excitation wavelengths of titanium ions. For this purpose, Figure 5 shows combined excitation spectra for the titanium-doped germanate glass sample (0.005 mol%). Upon excitation at 440 nm, the band at 270 nm might be identified as Ti ions in a tetravalent oxidation state [44,45]. On the other hand, our experimental investigations demonstrated the excitation band by monitoring the emission at 530 nm. The spectrum exhibited a relatively intense band at about 300 nm, which could be related to the Ti^3+^ ions [44, 45].

We performed the same procedure for the GT1, GT2, and GT3 glass samples (Figure 6). Obviously, changing the chemical glass composition leads to changes in the positions and intensities of the relevant excitation bands of the titanium species. Within the VIS range, the excitation spectra consisted of a characteristic band centered at 390 nm (Figure 6a). The relatively highest intensity of these excitation band glasses suggests a larger Ti^3+^ ions concentration in these samples. Notably, the band associated with titanium ions on the tetravalent valence state was very weakly detectable. Further analysis demonstrated notable changes in the excitation spectra (λ_em_ = 530 nm) dependent on the TiO_2_ content. To evaluate a full picture of the optical properties of the titanate–germanate glass system, we performed a luminescence experiment using the direct excitation wavelength between two excitation bands (λ_exc_ = 345 nm). 

The emission properties of GeO_2_-TiO_2_-BaO-Ga_2_O_3_ glass systems with different GeO_2_:TiO_2_ molar ratios were investigated under the 345 nm excitation wavelength and recorded in the 400–650 nm range, as shown in Figure 7. The asterisk (*) refers to the extremely low titanium dioxide concentration of the glass samples. For the fabricated titanate–germanate glasses (GT1, GT2, GT3), the intensities of the recorded band at 560 nm increased with the increasing GeO_2_:TiO_2_ molar ratio. Interestingly, for the sample with the highest GeO_2_:TiO_2_ molar ratio (1:5), the concentration quenching of the emission of the Ti^3+^ ions was observed. Our research phenomenally demonstrated that the successive replacement of germanium dioxide by titanium dioxide shifts the redox equilibrium Ti^3+^–Ti^4+^ to obtain optically detectable amounts of Ti^3+^ ions. Hence, a sample with a 10 mol% concentration of TiO_2_ is preferable for achieving a high luminescence emission of Ti^3+^ ions. Broadband, low-intensity emission with a maximum of 440 nm is characteristic of the Ti^4+^ valence state [44,45]. 

The absorption, excitation, and luminescence results showed that titanium dioxide caused variations in the valence state of the titanium ions in the germanate glass network that may produce structural modifications and local-field variations in the structure. For this reason, this work discusses the IR and Raman spectroscopy of germanate-based glass systems modified by TiO_2_/B_2_O_3_.

### 3.2. Raman and FT-IR Spectroscopy of Barium Gallo-Germanate Glasses Containing TiO_2_/B_2_O_3_

Previous reports showed that many types of germanate glass had been studied using vibrational spectroscopy [46,47,48,49]. Initially, the precise determination of the network units characterized by multicomponent germanate matrices was a complicated task due to the character of composed network-forming or network-modifier oxides. The network of germanate glass is formed by tetrahedral GeO_4_ structural units, which share their corners, and the Ge atom is covalently bonded to four bridging oxygens. The thermodynamic instability of the GeO_6_ octahedral units produces a large concentration of nonbridging oxygen ions. This evolution clearly indicates the conversion of [GeO_4_] → [GeO_6_] structural units [50]. According to the literature [51], the vibrational spectrum of the germanate glasses is rather remarkable. The germanate matrix is characterized by the structural units’ dominant contributions in low- and high-frequency regions. The low-frequency region is mainly characterized by a peak around 560 cm^−1^ and is associated with the bending vibrations of Ge-O-Ge. The high-frequency region is reported to contain a band at approx. 915 cm^−1^, and low inflections at approx. 1000 cm^−1^ are attributed to the asymmetric vibrations of the Ge-O-(Ge) bridges. These bands occur in a typical infrared spectrum when the glassy germanium oxide is composed of germanium–oxygen tetrahedra with nonbridging oxygens. It was repeatedly demonstrated that changes in the local structure of the germanate network as the alkali concentration [52,53] increased resulted in a systematic shift in the band components associated with the vibrations of the GeO_4_ units. This is evidenced by the broken Ge-O^-^ bridges at about 750 and 870 cm^−1^ for the Q2 and Q3 units. Comprehensive studies of germanate glasses have also provided strong evidence regarding the replacement of GeO_4_ by other units, including germanate–oxygen octahedra (GeO_6_). This strong modification is demonstrated by the appearance of a band at about 715 cm^−1^ in the midinfrared spectrum, evidently involving a change in the coordination number of the germanium ions from four (LK = 4) to six (LK = 6) [54,55]. According to the paper published by McKeaon and Marzbacher [56], when the GeO_2_ content decreases, the midfrequency envelope shifts to higher frequencies while the high-frequency features shift to lower frequencies. It was interpreted as a reduction in the average ring size, as well as an average lengthening of the T-O (where T is Ge or Ga) band. 

Figure 8 shows the measured FT-IR and Raman spectra in the wavelength range of 400–1600 cm^−1^ of barium gallo-germanate glasses with various GeO_2_:TiO_2_ molar ratios. The spectra exhibited two groups of bands (i) in the low-frequency region located from 400 to 600 cm^−1^ and (ii) in the high-frequency region from 700 to 900 cm^−1^. A single band dominates the first region of 400–600 cm^−1^ due to the GeO_4_ structural units, which share their corners, where the germanium atom bending is covalently bonded to four biding oxygens. The second high-frequency region, between 620–900 cm^−1^, is attributed to the GeO_6_ structural units, where the central atom is germanium and is surrounded by six oxygen atoms [57,58]. As expected, the molar ratio of TiO_2_ strongly affected these structural properties of glasses. As one can see from Figure 8, the intensities of the IR and Raman bands related to the GeO_4_ and GeO_6_ structural units underwent significant changes by incorporating the TiO_2_ content into the germanate glass host. With the introduction of TiO_2_ up to 30mol% in the GeO_2_-BaO-Ga_2_O_3_ glass network, the intensity band centered at about 450 cm^−1^ and 800 cm^−1^ was observed to significantly decrease with the shifting towards the lower frequency region. However, when the TiO_2_ concentration was greater than 30mol%, it was observed that the band due to the GeO_6_ structural units was shifted to higher frequencies. The explanation for these results lies in the dual role of titanium dioxide in the glass network. Titanium dioxide acted both as the network modifier and network former, which participated during the formation of the glass network in the form of TiO_4_ or existed in the gap outside the network in the form of TiO_6_ units [57,58,59,60]. In this work, the effect observed with the increased TiO_2_ content very well confirms that the doping titanium ions of the germanate matrix generated a strong destruction of germanate tetrahedra and octahedra units caused by the formation of more Ti-O structural units. We considered the Raman spectrum for the GeO_2_:TiO_2_ = 1:5 (GT3) glass sample when titanium dioxide acted as a network former. The main problem with this material is the interpretation of the local structure due to the overlapping bands. The most interesting observation concerning all registered spectra was the presence of a band in the frequency range of 600–700 cm^−1^. The mentioned band was very detectable with the increase in the amount of titanium ions introduced at the expense of germanium ions, which resulted in a systematic decrease in the amount of GeO_4_ tetrahedrons and GeO_6_ octahedrons. According to the data, this Raman peak is considered strong evidence of Ti-O stretching vibration connected with the TiO_6_ unit. Earlier studies of other glasses containing TiO_2_ exhibited a well-resolved band at about ~720 cm^−1^ identified due to the vibration of TiO_4_ structural units [61,62]. 

As a final point of our investigations, we characterized the structural properties of the barium gallo-germanate glass host containing B_2_O_3_ (Figure 9). As expected, independently, the chemical composition germanate glass structure resulted in the appearance of the spectra, revealing signals from the bending and stretching modes of the GeO_4_ and GeO_6_ structural units. In contrast to TiO_2_, the addition of boron oxide was found to be a weak scatterer in low-frequency and high-frequency ranges between 400–1600 cm^−1^. Adding B_2_O_3_ to glass causes progressive changes in the low and higher frequency range. These changes are accompanied by the decreasing of a strong band at 450 cm^−1^ (almost vanishing at 50 mol% B_2_O_3_ in the FT-IR spectrum) and the one at 800 cm^−1^ with increasing borate content and shifts to lower frequencies. The obtained results indicated that the Raman shift decreased from 1411 cm^−1^ (glass GeO_2_-rich composition) to 1340 cm^−1^ (glass B_2_O_3_-rich composition). A clear correspondence was observed between the bands in the Raman spectra and the measurements carried out of the infrared spectra for the obtained glasses. In general, in the boron-based glass host, boron is three-coordinated. Moreover, previous studies indicated the presence of trigonal and tetrahedral boron with different ratios and the partial conversion of BO_3_ into BO_4_ units [63,64].

To better visualize the obtained data, the deconvoluted components infrared bands were separated by a Gaussian deconvolution constructed for each sample GB1, GB2, and GB3 (see Figure 10). From the structural point of view, the most important observation was the change in the shift and the relative intensities of these two deconvoluted components positioned at 1230 cm^−1^ and 1360 cm^−1^ initiated as a function of the quantitative GeO_2_/B_2_O_3_ ratio. Hence, in assessing the effect of boron oxide on the barium gallo-germanate glass network, it is useful to analyze the location and mutual intensity of the appropriate deconvoluted bands, which have been previously described in detail in the scientific report of lead-borate glass systems [65].

The peculiar structural properties of borate glasses come from the ability of boron to occur in three or four coordination. In general, the equilibrium of the structural conversion between the BO_3_ and BO_4_ units in the glass network depends on the chemical composition and the kind of modifiers [66,67]. Moreover, the relative fraction of the BO_3_ and BO_4_ groups proved to be a sensitive probe of the basic structural units of the network. Calculations were evaluated with the following formula presented in Figure 11, where A_3_ and A_4_ correspond the areas of the BO_3_ and BO_4_ groups. The relative integrated intensity A3/A4 of the GeO_2_-B_2_O_3_-BaO-Ga_2_O_3_ glass system upon adding oxides GeO_2_ and B_2_O_3_ in three different concentrations of 10%, 30%, and 50% each drastically reduced from 1.03 (glass sample GB1) to 0.38 (glass sample GB3), respectively. We can conclude that the incorporation of B_2_O_3_ changed the local structure of the barium gallo-germanate glasses, and the low polymeric states containing the BO_3_ units constantly transformed into the high polymerized network units that mainly consisted of BO_4_. Interestingly, the group Mogus-Milankovic et al. [68] noted results for the quaternary Li_2_O-B_2_O_3_-P_2_O_5_-Ge_2_O glass system. With the addition of lower GeO_2_ content, the dominant borate unit was BO_4_, whose charge was more delocalized and enhanced the ion transport. In contrast, the formation of neutral BO_3_ units at a higher GeO_2_ concentration could break the conduction pathways and reduce the mobility of the ions. According to the literature [69,70,71], the competition for charge compensation between highly charged cations is the fundamental reason for boron coordination. The field strength of different modifier cations such as Na^+^, Ba^2+^, and Ca^2+^ [72] affects the enhanced stabilization of tetraborate groups by involving higher field strength cations in NBO-rich glasses. A similar effect of Na^+^ in sodium borate glasses has been recently found for the highly charged state of lanthanide ions (which act as modifiers) [73], which boosts the formation of negatively charged tetrahedral boron.

## 4. Discussion

The properties of precursor glasses were characterized using various experimental techniques to fulfill laser sources emitting midinfrared radiation and broadband optical amplifiers operating at near-IR range requirements. The obtained novel titanate–germanate and borogermanate glasses with a TiO_2_ and B_2_O_3_ content of up to 50 mol% were transparent and exhibited a fully amorphous nature. The luminescence studies of the GeO_2_-TiO_2_-BaO-Ga_2_O_3_ glasses perfectly confirmed the oxidation state of titanium ions in the function of the GeO_2_:TiO_2_ molar ratio. In the case of the GT series of glasses, the titanium acted as a glass network and a network modifier in the barium gallo-germanate network. It was proven that the incorporation of two various metal oxides in the barium-gallo germanate glass host influenced their structural GeO_4_ and GeO_6_ units. Indeed, this effort led to the better identification of the structural building blocks and the evolution of the Raman and FT-IR spectra of the BaO-Ga_2_O_3_-GeO_2_ glass system containing TiO_2_/B_2_O_3_ in the lower and higher frequency, respectively. It is well known that in glasses doped with lanthanide ions, the highest-energy phonons exercise the most influence in nonradiative relaxations because multiphonon decay occurs with the fewest number of phonons required to bridge the energy gap between two manifolds. On the other hand, the lower phonon energy of the glass host can reduce the probability of nonradiative relaxation and enable the higher quantum efficiency of photoluminescence and/or a higher luminescence lifetime of the excited state of lanthanide ions [74,75]. Moreover, we indicated [76] that the highest phonon energy for glass with GeO_2_:TiO_2_ equal to 1:1 decreased to 765 cm^−1^, which was smaller than that of the pure barium gallo-germanate glass reported (845 cm^−1^) [77]. For this reason, in the field of developing the glass structure system by embedding optically active ions to enhance their optical characteristics, titanium dioxide is a very useful component. Our systematic investigations demonstrated that barium gallo-germanate glasses containing TiO_2_ can be successfully used for near-IR laser applications at 1.06 µm through Nd^3+^ doping. The same glass systems doped with Er^3+^ ions are suitable for near-IR luminescence at 1.5 µm and could be useful for near-infrared broadband optical amplifiers. However, the optimal molar ratios of GeO_2_:TiO_2_ in these glass systems were completely different for Nd^3+^ [78] than Er^3+^ [79] ions. The further characterization of barium gallo-germanate glasses with TiO_2_ and their energy transfer processes between Yb^3+^ and Ln^3+^ ions (Ln = Pr, Er, Tm, Ho) [80] allowed us to demonstrate that the measured lifetimes decreased with an increasing TiO_2_ content, while changes in the energy transfer efficiency seemed to be nonlinear. To extend the optical applications of barium gallo-germanate glasses, the effect of B_2_O_3_ on structural modifications in the higher frequency range was observed, suggesting the important role of the molar ratio of GeO_2_:B_2_O_3_ in the formation of the glass host. In general, borate glasses are striking hosts because they are highly transparent, thermally stable, and show an appreciable solubility of lanthanide ions [81]. In the part of this work devoted to the results the Raman band near 1300 cm^−1^ was very well observed after B_2_O_3_ incorporation. This Raman band was related to the maximal phonon energy of the borate glass host, which caused emission quenching and the suppression of radiative emissions, especially in the near-IR spectral range. For that reason, barium gallo-germanate glasses containing relatively higher B_2_O_3_ concentrations are rather useless for near-IR luminescence applications, but they are interesting glass materials for the emission of visible light. These glass systems doped with Dy^3+^ ions emitted an intense greenish light, which was changed to yellowish light with an increasing GeO_2_:B_2_O_3_ molar ratio [82]. Interestingly, increasing B_2_O_3_ concentrations in barium gallo-germanate glasses did not negatively affect the reddish–orange luminescence and experimental lifetimes of Eu^3+^ ions despite their relatively high phonon energy [83]. This suggests evidently that barium gallo-germanate glass can be an excellent candidate for visible or near-IR luminescence depending on the glass modifiers (TiO_2_ or B_2_O_3_) and lanthanide doping. 

## 5. Conclusions

Undoped germanate-based glasses modified by TiO_2_/B_2_O_3_ were studied experimentally using XRD, luminescence, FT-IR, and Raman spectroscopy. The results led to the following conclusions: Barium gallo-germanate glass hosts can accommodate 50 mol% TiO_2_ and B_2_O_3_, and the samples were still fully amorphous. Based on the absorption spectra measurements, the absorption edge was determined. It was proven that the intensities of the excitation and emissions and the position bands of Ti^3+^ and Ti^4+^ strongly depended on the chemical composition of the fabricated materials.Analysis of Raman and FT-IR spectra for the modified barium gallo-germanate glasses showed evidence of GeO_4_ and GeO_6_ structural units, independently of the GeO_2_/TiO_2_ and GeO_2_/B_2_O_3_ molar ratio. However, titanium dioxide strongly modified the structure of the glass between the 400 cm^−1^ and 1000 cm^−1^ frequency region, while boron trioxide modified the structure between the 1100 cm^−1^ and 1600 cm^−1^ frequency region. As the titanium dioxide increased, the bands were shifted to a lower frequency region. From the Raman spectra, we observed that the additional band located near 650 cm^−1^ confirmed the presence of the TiO_6_ unit. The dependence of the fractions of the BO_3_ and BO_4_ units on the kind of glass network formers was reduced from 1.03 to 0.38. Therefore, we confirmed such a hypothesis that there was a strict correlation between the local structure and optical properties of the barium gallo-germanate glass system in the function of two various network-former components (TiO_2_ and B_2_O_3_). The presented analysis confirmed that the developed materials are one of the most important classes of matrices for doping optically active ions for photonic applications.

## Figures and Tables

**Figure 1 materials-16-01516-f001:**
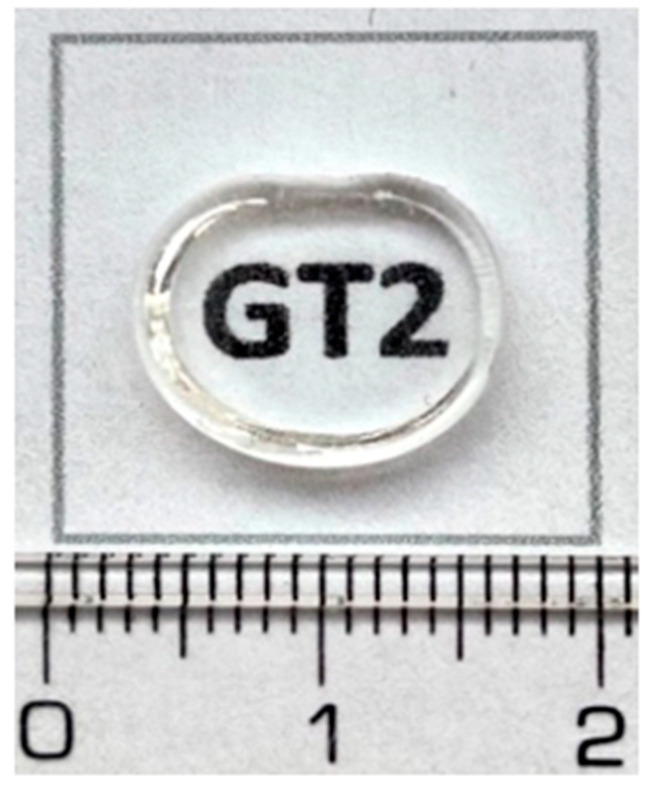
Photographs of the obtained glass sample: 30GeO_2_-30TiO_2_-30BaO-10Ga_2_O_3_ (GT2).

**Figure 2 materials-16-01516-f002:**
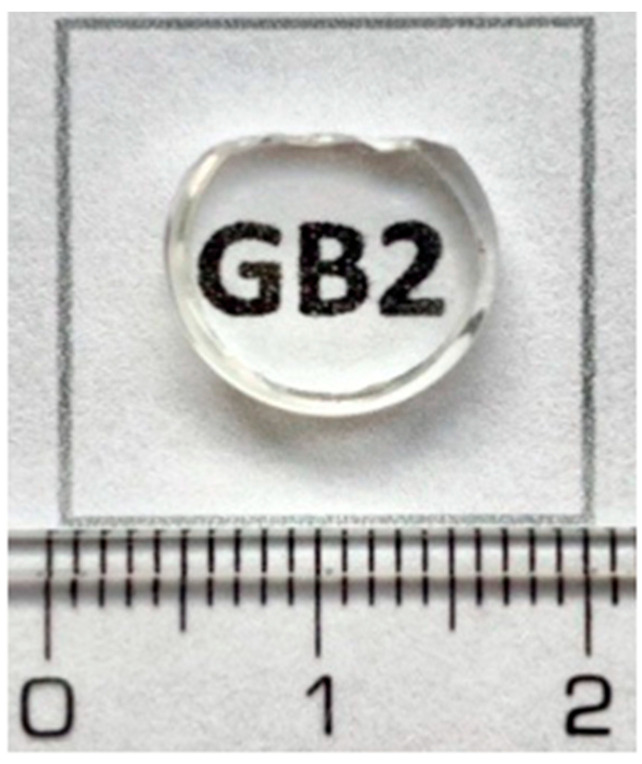
Photographs of the obtained glass sample: 30GeO_2_-30B_2_O_3_-30BaO-10Ga_2_O_3_ (GB2).

**Figure 3 materials-16-01516-f003:**
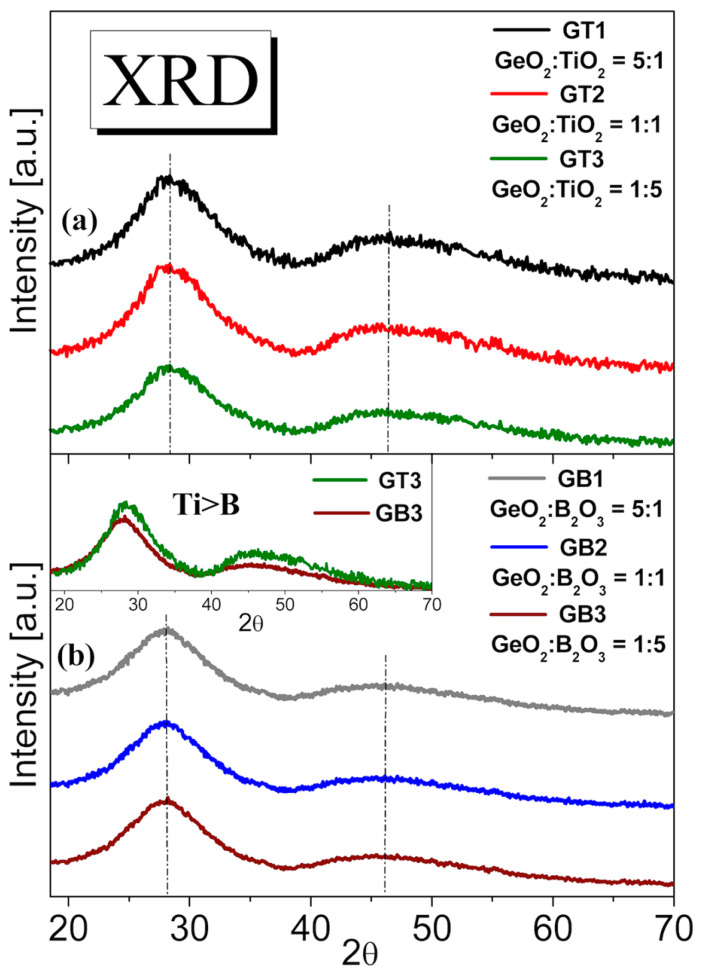
XRD profiles of barium gallo-germanate glasses containing various molar ratios TiO_2_ (**a**) and B_2_O_3_ (**b**). Insets show the X-ray diffraction intensity for GT3 (GeO_2_:TiO_2_ = 1:5) and GB3 (GeO_2_:B_2_O_3_ = 1:5) glass samples.

**Figure 4 materials-16-01516-f004:**
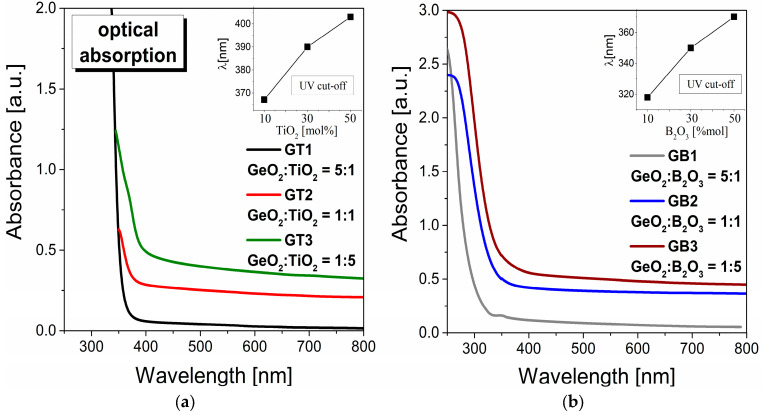
UV–visible optical absorption spectra for glass systems modified by TiO_2_ (**a**) and B_2_O_3_ (**b**). Insets show the variation in UV cut-off as a function of TiO_2_ and B_2_O_3_ concentration.

**Figure 5 materials-16-01516-f005:**
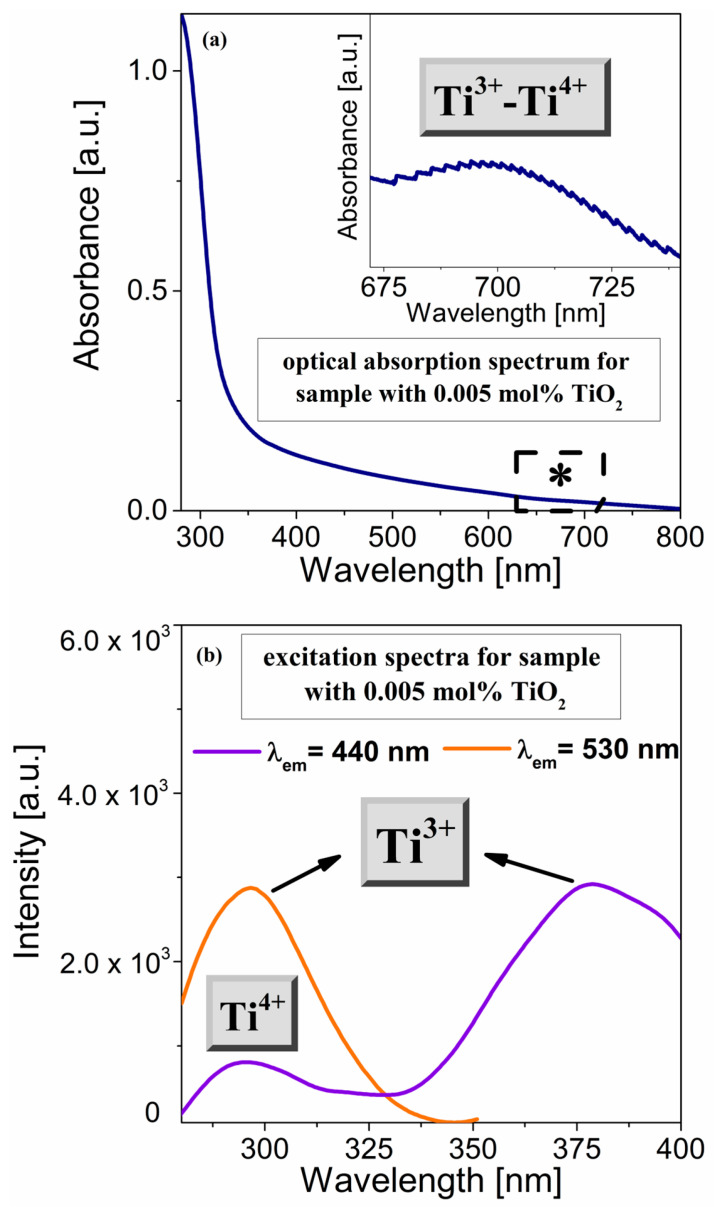
Optical absorption spectrum (**a**) and excitation spectra (**b**) of the sample containing extremely low titanium dioxide content. The excitation spectra are monitored at 440 nm and 530 nm. The inset shows Ti^3+^–Ti^4+^ pair’s interaction in the range 675–730 nm (*) recorded with high resolution.

**Figure 6 materials-16-01516-f006:**
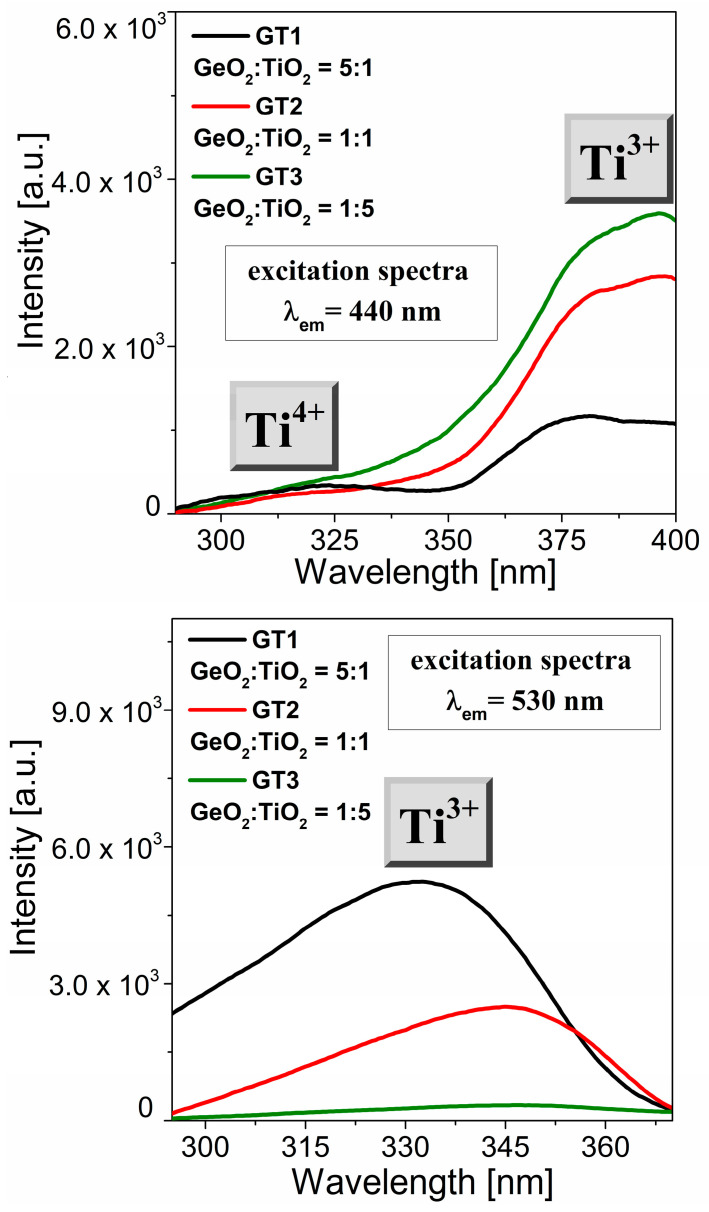
Excitation spectra (λ_em_ = 440 nm and λ_em_ = 530 nm) for GT1, GT2, GT3 samples.

**Figure 7 materials-16-01516-f007:**
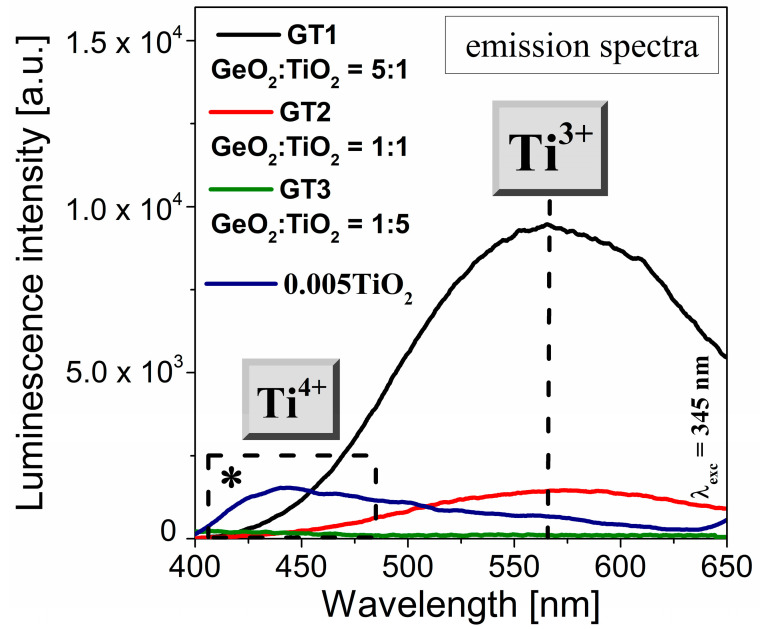
Visible emission spectra (λ_exc_ = 345 nm) for GT1, GT2, and GT3 samples. The asterisk (*) refers to glass containing an extremely low titanium dioxide concentration.

**Figure 8 materials-16-01516-f008:**
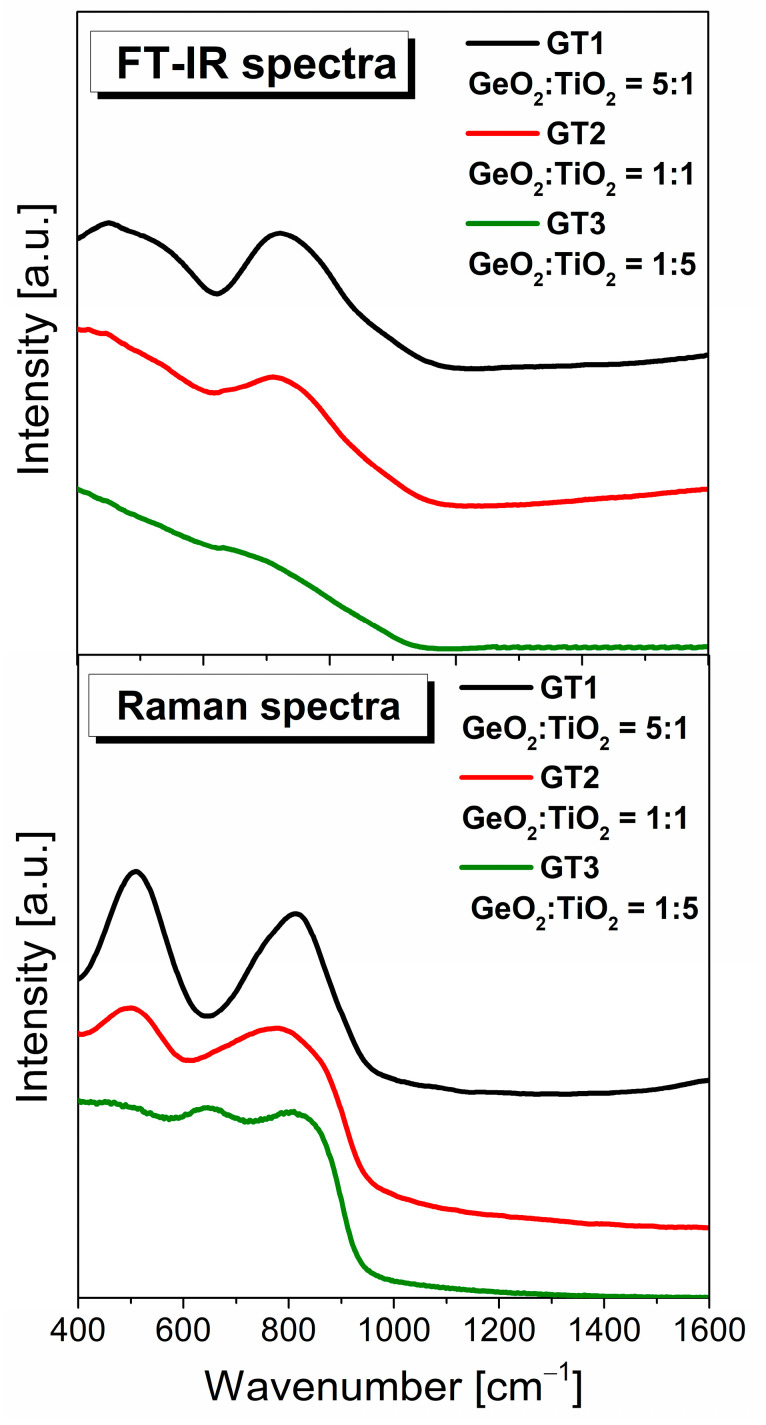
Infrared and Raman spectra of the investigated samples with GeO_2_/TiO_2_ ratio (5:1, 1:1, and 1:5).

**Figure 9 materials-16-01516-f009:**
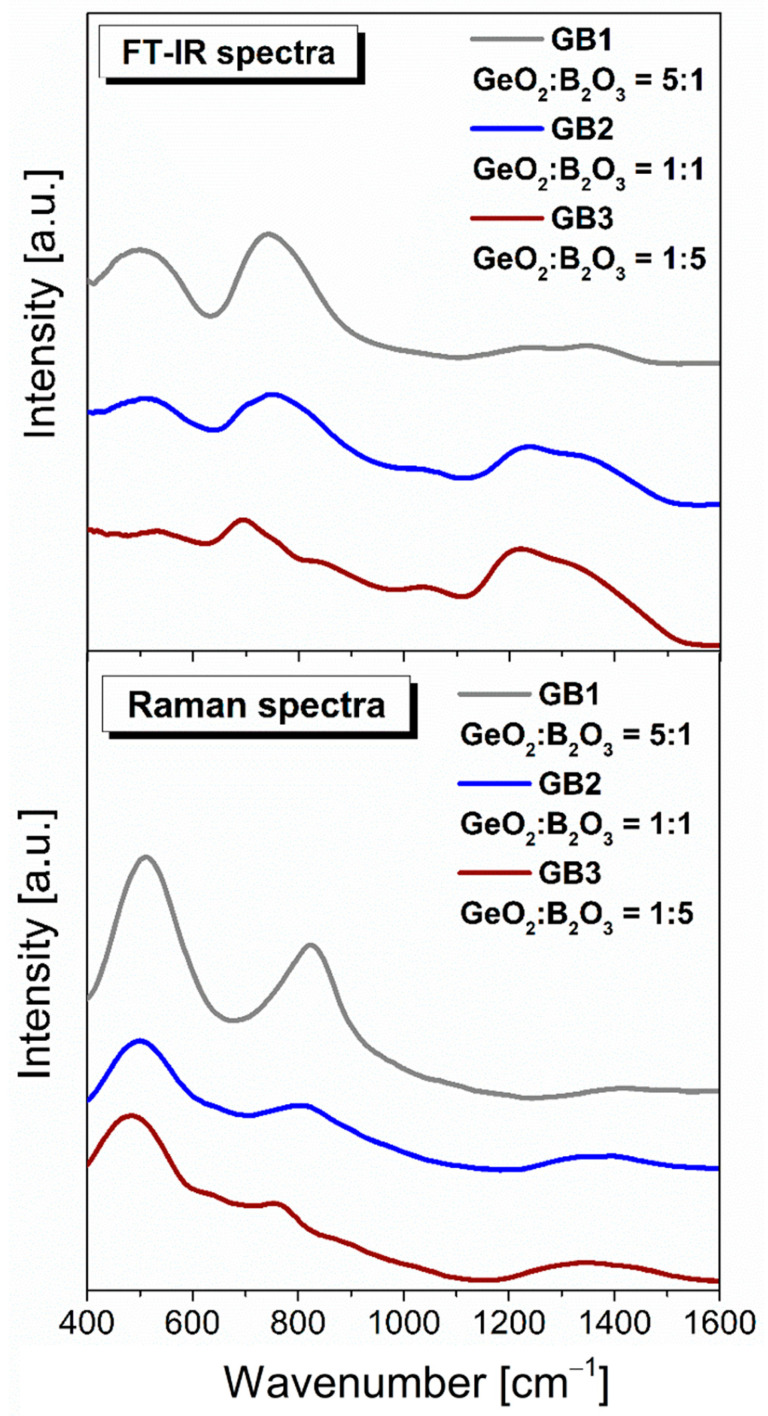
Infrared and Raman spectra of investigated glass samples with GeO_2_/B_2_O_3_ ratio (5:1, 1:1, and 1:5).

**Figure 10 materials-16-01516-f010:**
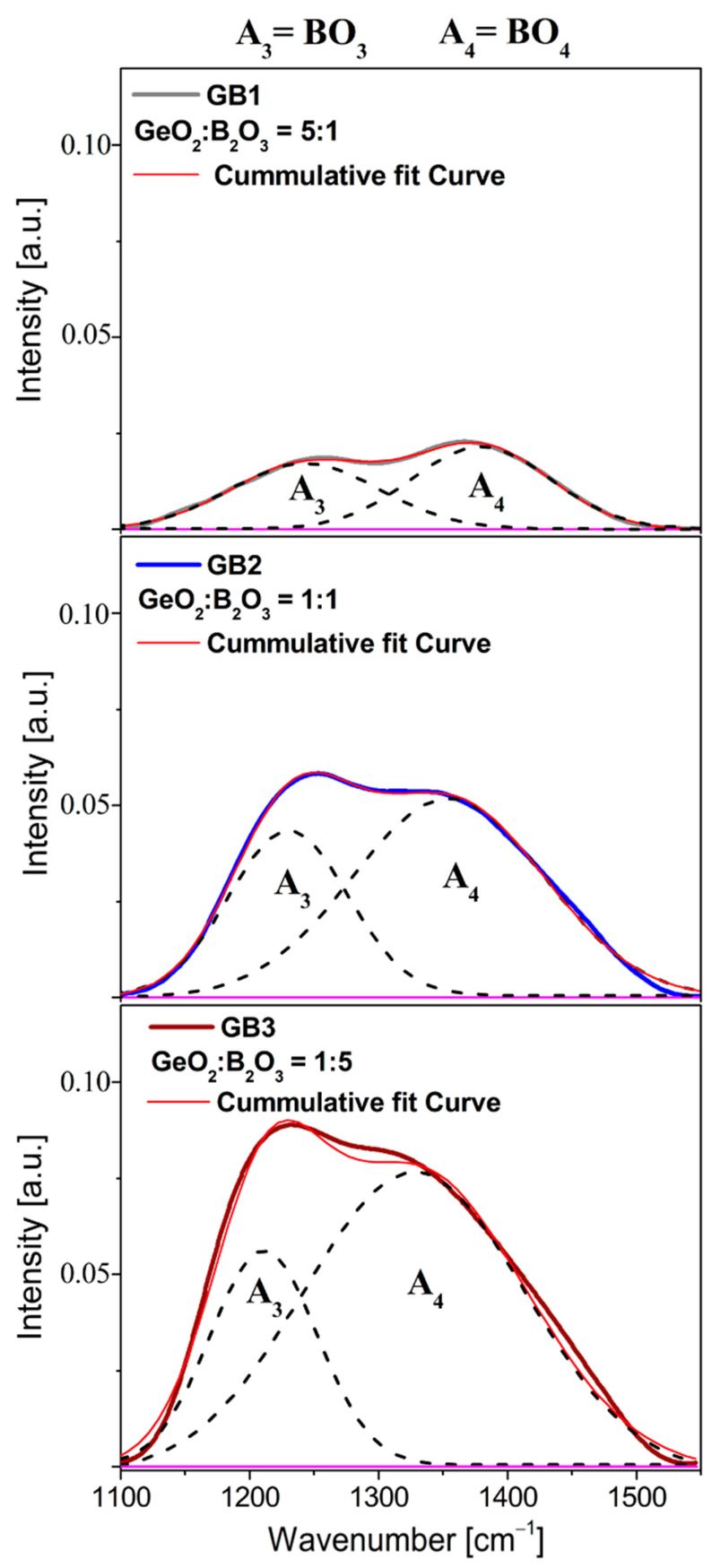
Deconvoluted infrared bands of series glass samples (GB1, GB2, GB3) as a function of GeO_2_ and B_2_O_3_ concentration in spectra range 1100–1550 cm^−1^.

**Figure 11 materials-16-01516-f011:**
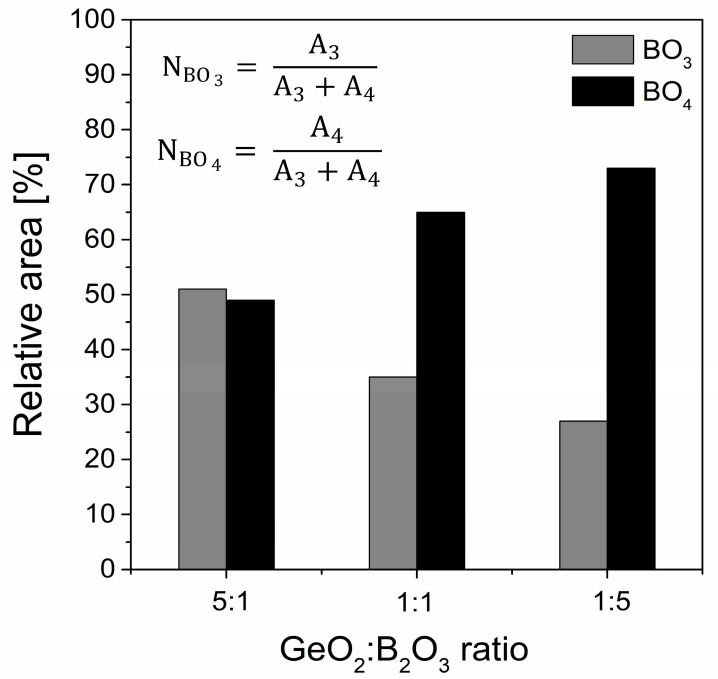
Change with composition of the calculated total fraction of BO_3_ and BO_4_ units versus the GeO_2_/B_2_O_3_ molar ratio.

**Table 1 materials-16-01516-t001:** Nominal composition (mol%) and GeO_2_/TiO_2_ ratio of glass samples.

Chemical Composition of BGG Glass with TiO_2_ (mol%)					
Sample Code	GeO_2_	TiO_2_	BaO	Ga_2_O_3_	GeO_2_:TiO_2_
GT1	50	10	30	10	5:1
GT2	30	30	30	10	1:1
GT3	10	50	30	10	1:5

**Table 2 materials-16-01516-t002:** Nominal composition (mol%) and GeO_2_/B_2_O_3_ ratio of glass samples.

Chemical Composition of BGG Glass with B_2_O_3_ (mol%)					
Sample Code	GeO_2_	B_2_O_3_	BaO	Ga_2_O_3_	GeO_2_:B_2_O_3_
GB1	50	10	30	10	5:1
GB2	30	30	30	10	1:1
GB3	10	50	30	10	1:5

## Data Availability

Not applicable.

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
