# Peer review of "Raman and Infrared Spectroscopy of Barium-Gallo Germanate Glasses Containing B2O3/TiO2"

_materials, 2023, doi:10.3390/ma16041516_

Round 1

Reviewer 1 Report

Dear Editor,

The manuscript by Karolina Kowalska et al. deals with the study of Ba-Ga-Ge glasses modified by substituting B2O3 or TiO2 for GeO2.

The two glass series have been investigated via vibrational spectroscopy techniques (Raman and ATR-FTIR) and optical absorption.

The major problems are the lack of real motivation for this study and the various sentences reporting results not shown in the current paper. For example, in the abstract, the last sentence reports:  "It was demonstrated that modified BGG glasses are an excellent choice for use in application as a laser gain active media operating in the NIR range."

However, this paper does not deal with such applications.

In the latest section (3.3), the authors report a review of THEIR published work using the same glass systems. Since it is just a list of already published data, both the title "potential optical applications" and the sentence in the abstract are misleading.

More appropriate data should be reported if the paper focuses on structural variations along with the measured compositions.

Based on their assertion, "A variety of experimental techniques have been successfully used to determine the structure of fabricated glasses, among others, X-ray diffraction (XRD) to confirm the amorphous nature of the glass samples, UV-visible optical absorption to spectroscopic assignments of titanium ions. As part of the contribution to research and knowledge available on structural characterization, the glass system was done using Fourier transform infrared (FTIR) measurements. Furthermore, Raman spectroscopy has been investigated to establish a composition/structure relationship. Another aspect worth exploring was a mathematical procedure of spectral decomposition as the possibility of determining the component bands resulting from matrix vibrations for [...]"

UV-Vis data: "to spectroscopic assignments of titanium ions." The data presented are really basic. No real assignment or quantification is provided.

All analyses are superficial, and sentences like "This shifting of the absorption edge occurs due to the creation of the NBO’s inside the studied glass sample."(lines 224-225) are not supported by any other data, or theoretical calculations of the bridging vs. non-bridging oxygens.

How was the UV edge cut-off determined? Why a more detailed study with the relationship between Urbach energy and Ti/B content was not reported? No indication of the refractive index/ band gap / density / molar volume / polarizability are reported. These results could have been used to define IF a variation of the Ti coordination/average oxidation state was occurring or IF the edge shift is purely related to NBO.

Some sentences are also obscure (e.g., lines 219-220) "There was one absorption edge occurred in these barium gallo-germanate glasses."

The vibrational spectroscopic data shown are limited to the 400-1600 cm-1 frequency range. No details on data reduction are provided (signal corrections, background, normalization). The vibrations are not really discerned and all the section is poorly written. No data analysis is done. The deconvolution of the ATR-FTIR is debatable since no details on the data reduction are provided.

No real structure-relationship is reported. The data are never linked. Furthermore, the limited conclusions drawn from the optical data (based on the increase of NBO) are inconsistent with the increase of 4-fold coordinated B units reported in Figure 6.

I will not recommend the present paper for publication.

Author Response

My answer:

Thank you for your decision about our paper describing the composition/structure/property relationship barium gallo-germanate containing TiO2 and B2O3. We are sorry that the work didn't meet your expectations. We encourage you to read the revised paper version, whose scope has been expanded to include luminescence studies of fabricated glasses.

In the revised version of the paper, we paid special attention to the motivation of the presented research. It was generally accepted that rare-earth doped GeO2 are considered as promising optical materials for tunable broadband optical amplifiers and laser sources emitting infrared radiation. Indeed, the last section of this paper, based on our previous research, confirms that titanium dioxide and boron oxide have a positive effect on the luminescent properties of barium gallo-germanate glasses doped with rare-earth ions. To the best of our knowledge, GeO2-TiO2-BaO-Ga2O3 and GeO2-B2O3-BaO-Ga2O3 glass host matrices without optically active ions have not been analyzed using Raman and infrared spectroscopy.

The UV cut-off wavelength is defined as the intersection between the zero base line and the extrapolation of the absorption edge. Analysis of the optical absorption spectra was to demonstrate how the absorption edge shifts as a function of the content of the two glass-forming oxides. We completely agree with the Reviewer's that the the relationship between Urbach energy and the refractive index/ band gap / density / molar volume / polarizability would form the basis for defining if a variation of the Ti coordination/average oxidation state was occurring or if the edge shift is purely related to NBO. This suggestion will certainly be considered in further detailed studies of the developed glass host matrices.

We agree with the Reviewer that the current form of the Infrared and Raman results and their analysis should be improved by experimental and signal processing data. The IR and Raman spectra were recorded in the full range (400-4000 cm-1); above 1600 cm-1, there were no significant and intensity vibrational bands. Especially, the band's intensity in the 3000-3800 cm-1 frequency region, which corresponds to hydroxyl groups stretching vibrations, was almost completely reduced for all samples indicating the elimination of hydroxyl groups of host matrices. Therefore, the vibrational spectroscopic data were limited to the 400-1600 cm-1 frequency range due to analyzing the characteristic vibrations corresponding to germanate and titanate structural units. A total 128 scans for Infrared spectra with a resolution of 4 cm-1 were averaged for each sample. A total 120 scans for Raman spectra were registered. The final IR and Raman spectra of the studied samples were obtained by subtracting the background and applying baseline correction to separate proper spectroscopic signals from interference. Then the registered data have been pre-proceed (signal processing: smoothing) using Origin Pro software. Moreover, in order to better characterize the influence of glass composition on structural properties, Raman's spectra were deconvoluted using Origin Pro software. The coefficient of determination (R square) of all the deconvoluted Raman spectra was 0.99.

Some sentences together were added to the revised version of the manuscript.

We would like to thank the Reviewer for comments and the favorable review of our article once again. We believe that thanks to the corrections introduced the manuscript in the amended version is worthy of publication in Materials.

Reviewer 2 Report

This article presents a study of preparation and structural-optical characterization of BaO-Ga2O3 based glasses with B2O3/TiO2.

The idea seems interesting. However, way of listing without linking them in one clear story results in decrease quality. My suggestion is to change it and make better connections between different parts in the manuscript (results and discussion) which will result in a better outcome. The language and writing style need to be work on. In the light of it, my suggestion is revision before further consideration for publication. I have doubts is this manuscript in this form and lack of additional characterization and data, appropriate for this Journal.

In the light of scientific content, there are some points that I feel should be addressed and clarified:

1.         Abstract and Conclusion: Please make it more condensed with emphasis on obtained results and conclusions.

2.         Literature: Please, add up-to-date references where possible, and explain in more detail the idea behind this investigation.

3.         Results and discussion: Please, place additional emphasis on optical properties and correlation to structural ones.

4.         Results and discussion: PXRD: the composition is drastically changing, so it would be expected to see difference in intensity and slight shift in amorphous humps. Here, it can be seen no changes if there is change in composition from dominantly germanium to borate/TiO2 glasses.

5.         Results and discussion: Raman and FTIR data: what about first GT series and deconvolution of spectra. There is strong cut off around 900-950 in Raman and 1000 cm-1 in FTIR spectra. Maybe it is a matter of background? It is not visible in GB series.

6. Only 6 Figures. Please think on the additional experiments which will complement this study.

6.         I also suggest editing of English language and style.

Author Response

My answer:

First of all, we would like to thank you very much for the valuable comments necessary to improve our manuscript entitled “Raman and Infrared Spectroscopy of Barium-Gallo Germanate Glasses Containing B2O3/TiO2”.

  1. Abstract and Conclusion: Please make it more condensed with emphasis on obtained results and conclusions.

My answer

According to Your suggestion we created an abstract and conclusions that highlight the key results and recommendations of the fabricated barium gallo-germanate glasses containing two various glass network-formers (TiO2 and B2O3).

  1. Literature: Please, add up-to-date references where possible, and explain in more detail the idea behind this investigation.

My answer

According to Your recommendation, some additional up-to-date references concerning the structural properties of various composition germanate-based glasses have been added to the revised version of the manuscript. Barium gallo-germanate glasses have been intensively studied as a promising candidate in optical applications. Experimental data indicate that titanium dioxide can positively affect the optical properties of glasses. However, some issues, such as the crystallization of titanium dioxide, may have impeded the fabrication of high-quality glass host matrices. What is interesting, our previous studies confirmed that two different glass-forming oxides show different effects on the luminescence profile of transition metal ions. However, in this case, to demonstrate changes in the local environment, GeO2-B2O3-BaO-Ga2O3 and GeO2-TiO2-BaO-Ga2O3, trivalent chromium ions were used as an extremely useful spectroscopic probe. In the presented work, we have shown that the precursor glass shows a fully amorphous character. Moreover, we analyzed the properties of undoped matrices as a function of GeO2:TiO2 and GeO2:B2O3 content using two complementary research tools such as Raman and infrared spectroscopies. The novelty of the studied glass systems was specified in the revised version of the manuscript.

  1. Results and discussion: Please, place additional emphasis on optical properties and correlation to structural ones.

My answer:

According to the Reviewer’s suggestions, some sentences were modified in Results and Discussion section to indicate the emphasis on optical properties and correlation to structural ones.

  1. Results and discussion: PXRD: the composition is drastically changing, so it would be expected to see difference in intensity and the slight shift in amorphous humps. Here, it can be seen no changes if there is change in composition from dominantly germanium to borate/TiO2.

My answer

We conducted the X-ray diffraction patterns for each glass sample (GT and BG series) again to confirm the amorphous nature of the glass samples. In both cases, for the fabricated multicomponent glass systems, the obtained results are reproducible, and we don’t observe relationships related to the change of shift as a function of TiO2 content (GT1, GT2, GT3 glass series) and B2O3 (GB1, GB2, GB3 glass series). On the other hand, the inset in Figure 3 in the present Revised manuscript, the change in the intensity of representative X-ray diffraction patterns is very well observed. The GT3 glass sample (50 mol% TiO2) shows a higher peak intensity than the glass sample containing GB3 (50 mol% B2O3), which meets the requirement that the higher the atomic number of an element (Ti>B), the higher the X-ray diffraction intensity.

  1. Results and discussion: Raman and FTIR data: what about first GT series and deconvolution of spectra. There is a strong cut-off around 900-950 in Raman and 1000 cm-1 in FTIR spectra. Maybe it is a matter of background? It is not visible in GB series.

My answer:

The IR and Raman spectra were recorded between 400 cm−1 and 1000 cm−1 frequency region consisting of two main bands centered at about 500 cm−1 and 800 cm−1. It clearly indicated that titanium dioxide strongly affects the destruction of structural units [GeO4] and [GeO6]. The Raman spectra very well evidenced band near 650 cm−1 corresponding to the stretching vibration of Ti-O in TiO6 unit. Above 1000 cm-1, there were no significant and intense vibrational bands. In the case of the spectra for the titanate-germanate series, deconvolution was not done because, so far in the literature, no one has analyzed structural changes in this way. It should be noted the deconvolution procedure and the analysis of the infrared bands have been described in detail and discussed for borate systems in numerous works. The presented research confirmed that this is a well-described method that allows the correct interpretation of the data obtained. The final IR and Raman spectra of the studied samples were obtained by subtracting the background and applying baseline correction to separate proper spectroscopic signals from interference. Then the registered data have been pre-proceed (signal processing: smoothing) using Origin Pro software.

  1. Only 6 Figures. Please think on the additional experiments which will complement this study.

My answer

According to the Reviewer’s suggestions, the optical properties of fabricated glass containing titanium dioxide were also confirmed by the luminescence spectra measurements under monitoring excitation wavelength at 345 nm. Based on literature data and our experiments, it is clearly seen that the broadband band centered at 560 nm corresponds to titanium ions at a trivalent oxidation state. These results are explained and discussed in the revised manuscript. The appropriately cited works were also given.

  1. I also suggest editing of English language and style.

According to the Reviewer`s suggestion, this slips of the pen along the manuscript were corrected.

We would like to thank the Reviewer for comments and the favorable review of our article once again. We believe that thanks to the corrections introduced the manuscript in the amended version is worthy of publication in Materials.

Reviewer 3 Report

In my opinion the article is interesting and worth to be published. I think that the obtained results are clearly presented, and their interpretation is reasonable. The conclusions are well supported by the experimental results. I recommend the acceptance of this article for publication in the present form.

Author Response

My answer:

The Authors would like to thank the Reviewer for reviewing our paper and for positive evaluation of our work. We would like to express sense of gratitude for appreciating our presented paper.

Round 2

Reviewer 1 Report

Dear Editor,

After reading the revised version, I must say that the presentation quality and readability started to improve. I want to thank the authors for their effort. However, the concerns I previously had were not completely addressed. Still, no real structure-relationship is reported. It is a pity that the different results are never linked. The authors agreed on the importance of correlating the properties with several parameters. But they did not include any, and the answer is not satisfactory: " This suggestion will certainly be considered in further detailed studies of the developed glass host matrices."Wasn't it the goal of this paper?

They added new data that could be of great help if appropriately evaluated. Unfortunately, as before, no real relationship is sought. E.g.,: the PL data were not evaluated. Results are just presented and never discussed or linked to the network connectivity and the vibrational modes. Are the structural roles of the different Ti populations the same?

"Independently on the excitation wavelengths 345 nm, the spectrum for GT1, GT2, and GT3 consists of a luminescence band characteristic for titanium ions." The mechanisms responsible for the PL emission are not reported. Where are the PL excitation spectra? They could provide support in evaluating the absorption edge evolution.

The optical absorption spectrum for the "new" designed glass with low TiO2 is not reported. Neither the vibrational spectra. How does this "new" sample help to provide a structure-properties correlation?

"Broadband, low-intensity emission with a maximum of 440 nm is characteristic for Ti4+ ions". Some references here should be included. The luminescence mechanisms as well should be reported.

I regret to say that my opinion is not changed. In its present form, this manuscript cannot be accepted. Further work is needed to link the different results and create a picture of the network evolution and how it influences the properties.

Author Response

Dear Reviewer,

First of all, we would like to thank you very much for the valuable comments necessary to improve our manuscript entitled “Raman and Infrared Spectroscopy of Barium-Gallo Germanate Glasses Containing B2O3/TiO2”.

The main objective of the designed new glass sample containing 0.005 %mol of titanium dioxide was to demonstrate from luminescence spectra that titanium ions exist as Ti4+ ions in the germanate matrix. In contrast, we developed glasses containing higher concentrations of titanium dioxide (10, 20, and 30 %mol) when TiO2 plays the role as a glass-network modifier and titanium ions are present in the trivalent oxidation state. In the introduction to this paper, we will emphasize that titanium dioxide plays a dual role in the glass host structure, depending on its concentration. The hypothesis has been proven. Notably, as the Reviewer recommended, we conducted spectroscopic studies that contributed new information to this work. An optical absorption spectrum was recorded for a "new" glass sample containing 0.005 %mol TiO2. The absorption spectrum showed no evidence of Ti4+ ions. In the case analyzed, the absorption edge, which is significantly shifted toward longer wavelengths, is a complication in correctly interpreting the different forms of titanium ions. This study was extended to register the absorption spectrum with high resolution to demonstrate the coexistence of the Ti3+-Ti4+ pair's interaction. In the revised paper, we have expanded our research in accordance with the Reviewer's recommendations by recording the excitation spectra using two selected monitoring wavelengths (440 nm and 560 nm). Considering the Reviewer's question about structure-properties correlation, we demonstrated using optical spectroscopy that titanium ions occur as Ti3+ and Ti4+ depending on the concentration (as modifier/dopant).

Meanwhile, structural characterization using Raman and FTIR spectroscopy showed that titanium dioxide has a destructive effect on vibrations from germanium atoms. Notably, the Raman spectrum for the GT3 glass sample with predominantly TiO2 content showed the presence of [TiO6] structural units when the measurement of the emission spectrum showed a luminescence quenching phenomenon of Ti3+ ions. As mentioned above, titanium plays a dual role in the glass structure: as the network former in the form of tetrahedral TiO4 or as the network modifier in the form of octahedral TiO6. It should be noted that the published information of the Ti4+ coordination in the literature, though, seems contradictory in that some authors suggest the presence of 4- and 6- coordinated Ti4+ while others also suggest 5-coordinated entities. In summary, excitation and luminescence spectra showed that titanium ions in titanium-germanate glass up to 30 %mol are predominantly present as Ti3+ ions. The occurrence of stretching vibration of Ti-O in TiO6 unit was shown by Raman spectrum. We think that the presented new results of the optical properties of titanium-germanate glasses have created a view of the evolution of the oxidation state of titanium ions in the germanate matrix.

According to the Reviewer’s suggestions, some sentences were modified in the Results and Discussion section. We want to thank the Reviewer for valuable comments on our article once again. We believe that thanks to the corrections introduced, the manuscript in the amended version is worthy of publication in Materials.

Reviewer 2 Report

Dear Authors, thank you making improvements to your manuscript.

Author Response

The Authors would like to thank the Reviewer for reviewing and positive evaluation of our paper entitled “Raman and Infrared Spectroscopy of Barium-Gallo Germanate Glasses Containing B2O3/TiO2. We would like to express sense of gratitude for appreciating our presented paper.